# MiRNAs as Potential Regulators of Enthesis Healing: Findings in a Rodent Injury Model

**DOI:** 10.3390/ijms24108556

**Published:** 2023-05-10

**Authors:** Carlos Julio Peniche Silva, Rodolfo E. De La Vega, Joseph Panos, Virginie Joris, Christopher H. Evans, Elizabeth R. Balmayor, Martijn van Griensven

**Affiliations:** 1Cell Biology-Inspired Tissue Engineering, MERLN Institute for Technology-Inspired Regenerative Medicine, Maastricht University, 6229 ER Maastricht, The Netherlands; c.penichesilva@maastrichtuniversity.nl (C.J.P.S.); delavega.rodolfo@mayo.edu (R.E.D.L.V.); v.joris@maastrichtuniversity.nl (V.J.); 2Musculoskeletal Gene Therapy Laboratory, Rehabilitation Medicine Research Center, Mayo Clinic, Rochester, MN 55905, USA; panos.joseph@mayo.edu (J.P.); evans.christopher@mayo.edu (C.H.E.); erosadobalma@ukaachen.de (E.R.B.); 3Experimental Orthopaedics and Trauma Surgery, Department of Orthopaedic, Trauma, and Reconstructive Surgery, RWTH Aachen University Hospital, 52074 Aachen, Germany

**Keywords:** microRNA, enthesis, healing, collagens, mRNA targets

## Abstract

MicroRNAs (miRNAs) are short non-coding RNA sequences with the ability to inhibit the expression of a target mRNA at the post-transcriptional level, acting as modulators of both the degenerative and regenerative processes. Therefore, these molecules constitute a potential source of novel therapeutic tools. In this study, we investigated the miRNA expression profile that presented in enthesis tissue upon injury. For this, a rodent enthesis injury model was developed by creating a defect at a rat’s patellar enthesis. Following injury, explants were collected on days 1 (*n* = 10) and 10 (*n* = 10). Contra lateral samples (*n* = 10) were harvested to be used for normalization. The expression of miRNAs was investigated using a “Fibrosis” pathway-focused miScript qPCR array. Later, target prediction for the aberrantly expressed miRNAs was performed by means of the Ingenuity Pathway Analysis, and the expression of mRNA targets relevant for enthesis healing was confirmed using qPCRs. Additionally, the protein expression levels of collagens I, II, III, and X were investigated using Western blotting. The mRNA expression pattern of *EGR1*, *COL2A1*, *RUNX2*, *SMAD1*, and *SMAD3* in the injured samples indicated their possible regulation by their respective targeting miRNA, which included miR-16, -17, -100, -124, -133a, -155 and -182. Furthermore, the protein levels of collagens I and II were reduced directly after the injury (i.e., day 1) and increased 10 days post-injury, while collagens III and X showed the opposite pattern of expression.

## 1. Introduction

The enthesis is a complex interphase tissue, connecting tendons and ligaments to bones. It presents the opposite gradients of cellular composition, collagen alignment, and mineralization that are necessary for stress dissipation at the tendon-to-bone attachment site [1,2,3]. However, the complexity of the tissue makes the enthesis a highly challenging subject of study for tissue engineers.

Due to the physiology of the enthesis, this tissue is susceptible to sport-related injuries, overuse, and degeneration [4,5,6,7]. Upon injury, the healing process that is triggered at the enthesis usually yields scar tissue that resembles neither the morphology nor the mechanical properties of the native enthesis, increasing the risk of recurrent injury [1,8,9,10]. Hence, the development of effective tissue-engineering strategies to aid the regeneration and healing of injured entheses constitutes an unmet need [4].

To address this challenge, we aimed to characterize the early stages of healing in a rat’s injured patellar enthesis by focusing on the expression patterns of a set of fibrosis-related microRNAs (miRNAs).

MiRNAs are a highly interesting class of molecules. They are short, non-coding RNA sequences, usually 21–25 nucleotides in length and possessing the ability to target specific mRNA molecules [11,12,13]. MiRNAs are generally complementary to the 3′-UTR region of their mRNA target, where they bind, terminating translation, and thus repress protein synthesis [12]. MiRNAs exhibit diverse expression patterns, intervening in many developmental and physiological processes [13]. Additionally, an individual miRNA can have several mRNA targets, while one mRNA target can be regulated by several miRNAs [14]. Hence, a single miRNA can participate in the regulation of several biological processes.

Since the first miRNA, lin-4, was described in 1993, over two thousand new miRNAs have been identified [15,16]. In 2016, 60% of all human protein-coding genes were estimated to be post-transcriptionally regulated by miRNAs [16]. Furthermore, several miRNAs have been identified to play key roles in cancer, liver, heart, and musculoskeletal diseases, among other conditions [14,17,18,19,20].

More recently, the inhibitory effect of miRNAs on their mRNA targets has inspired the use of miRNA mimics and inhibitors to specifically tailor their regulatory effects to different pathways. In the field of cancer research, miRNA replacement therapy aims to increase or restore the expression of tumor-suppressing miRNAs that might be downregulated or deleted in cancer cells, while miRNA reduction therapy aims to decrease or inhibit the expression of oncogenic miRNAs [21,22,23]. A similar strategy is being followed to fight alcohol-associated liver disease, fatty liver, and drug-induced liver injury using miRNA mimics or inhibitors, depending on the desired effect over the expressing miRNA [24,25].

In the field of musculoskeletal research, recent reports indicate that miRNAs can also be used as epigenetic regulators to improve the process of healing injured tendons and cartilage, as well as to aid in the process of bone healing and remodeling upon fracture [19,26,27,28,29]. Among other roles, miR-29a has been reported to regulate the in vitro expression of collagen III in fibroblasts from patients with systemic sclerosis [30]. Consequently, the treatment of injured tendons in a horse model with a miR-29a mimic showed improved early tendon healing by reducing collagen III transcript levels without affecting the expression levels of collagen I [28]. In an osteoarthritic cartilage model, in vitro transfection of chondrocytes with a miR-148a precursor decreased the expression of collagen X while increasing the expression of collagen II, which indicated the potential therapeutic use of this miRNA to modulate hypertrophic differentiation within these cells [29]. Additionally, in patients with traumatic brain injury and concomitant fractures, injection of miR-26a mimics at the fracture site led to increased bone formation, possibly through regulation of the phosphatase and tensin homolog deleted on chromosome 10 (*PTEN*) [31].

The occurrence of fibrosis, scar formation, and tendon adhesion during the healing of injured tendon has been previously linked to the overexpression or inhibition of specific miRNAs [32]. Especially relevant for fibrosis and tendon adhesion are those miRNAs capable of regulating the TGFβ signaling pathway in tenocytes and fibroblasts [32,33]. The TGFβ pathway is key for tendon development. However, the overactivation of such pathway leads to fibrosis and tendon adhesion [33]. Two well-known examples of miRNAs that regulate the TGFβ signaling pathway are miR-21-5p and miR-29b. While miR-21-5p activates the TGFβ-1 pathway by inhibiting *SMAD7* in tenocytes and fibroblasts, miR-29b inhibits the same pathway by targeting *SMAD3* [34,35]. Yet, in the context of the tendon-to-bone enthesis, little is known about the role of fibrosis-related miRNAs during either the healing process or scar formation after injury.

By investigating the pattern expression of fibrosis-related miRNAs upon injury in an enthesis animal model, we aimed to identify miRNAs with therapeutic potential that might aid the process of enthesis regeneration and healing.

## 2. Results

All the animals used in the study survived the surgical creation of the patellar injury and remained otherwise healthy until the collection of the explants.

### 2.1. miRNA Expression

The miRNA expression in samples harvested at days 1 and 10 after injury was assessed using a qPCR array. We observed that 1 day after the creation of the injury, 28 miRNAs were dysregulated over twofold more in comparison to their expression in the native tissue (Figure 1). Of this total, six miRNAs were upregulated, and 22 miRNAs downregulated. Furthermore, at 10 days after enthesis injury, 52 miRNAs resulted in dysregulation beyond the twofold cut-off. Of these, only seven miRNAs were downregulated while 45 miRNAs were upregulated.

The identities of all miRNAs found to be aberrantly regulated were uploaded to the Ingenuity Pathway Analysis (IPA) software v012004. Target prediction was conducted according to those gene networks of significance to enthesis-associated pathways. Those miRNAs and respective mRNA targets that were associated with cartilage and tendon pathways were identified using the IPA software v012004, and are summarized in Figure 2. In addition, we included in our analysis the interactions between miRNAs and mRNA targets that had been reported in the literature to be relevant for the regulation of tendon or cartilage pathways but which were not predicted by the IPA. These interactions between the miRNAs and respective mRNA targets are indicated in Figure 2 with an asterisk (*). Given their importance in the context of enthesis injury and to subsequent healing, we have investigated the mRNA expression in the injured samples, as well as with the IPA-predicted mRNA targets.

Results obtained by the miScript qPCR array were validated with qPCR in each individual sample, and for each investigated time of observation (Figure 3). Of note, only aberrantly regulated miRNAs that were deemed relevant to enthesis injury according to the IPA analysis were selected for validation.

### 2.2. mRNA Expression

To investigate the potential effect of the dysregulated miRNAs in the injured samples, the expression levels of their enthesis-relevant mRNA targets were evaluated using qPCR in each sample at both time points after the injury (Figure 4).

A downregulation was observed in the mRNA levels of *COL3A1*, *SMAD1*, *SMAD3*, *RUNX2*, *MKX*, and *EGR1* for the samples harvested 1 day following the enthesis injury. At day 10 following the injury, *RUNX2*, *MKX*, *SMAD3*, and *EGR1* remained downregulated beyond the twofold cut-off mark, while *COL1A1*, *COL2A1*, *COL3A1*, and *ANKH* showed increased levels of mRNA expression.

### 2.3. Western Blot

The protein expression levels of collagen I, collagen II, collagen III, and collagen X were investigated using the Western blot procedure. The results were normalized against total protein content and are depicted in Figure 5. A decrease was observed in the presence of collagen I and collagen II, as was an increase in the protein-content of collagen III and collagen X immediately after enthesis injury (i.e., day 1). Interestingly, 10 days after enthesis injury, the opposite pattern of collagen content was observed, showing an increase in the amount of collagen I and collagen II, and a strong decrease in the amount of collagen III and collagen X (Figure 6).

### 2.4. Histology

#### 2.4.1. Safranin O, Alcian Blue and Masson Trichrome Staining

Samples from both the native tissue and the injured tissue were stained with safranin O, alcian blue, and Masson trichrome (Figure 7). The intensity of the safranin O staining was proportional to the proteoglycan content of the cartilage tissue. In addition to proteoglycans, alcian blue also stains for mucopolysaccharides and glycoproteins. Masson trichrome stains collagen fibers and muscle. The staining of the samples harvested at 1 day after enthesis injury allowed us to visualize the extent of the injury at the enthesis site, while the observation of the stained samples harvested at 10 days after the injury provided a great deal of information on the healing process of the enthesis, the occurrence of fibrosis, and the eventual ectopic ossification.

The safranin O staining of the samples harvested at 10 days after the injury revealed an enlarged tendon, with a clear distinction between a native-like tendon towards the medial part of the tendon and a fibrotic portion towards the lateral part of the tendon, which was positively stained for proteoglycans. In the fibrotic area, the collagen fibers showed poor alignment compared to the alignment present in the uninjured tissue (Figure 7a). Proteoglycan was not observed in the native tendon tissue. Additionally, in the same group of samples (i.e., 10 days after enthesis injury) a dense area of extracellular matrix (ECM) was visible that resembled cartilage at the tendon side of the enthesis, whereas, in the native tissue, this area contained no cartilage but instead comprised the parallel aligned collagen fibers characteristic of the tendon tissue.

Similarly, the alcian blue and Masson trichrome staining of the samples harvested 10 days after the injury showed positive blue staining in the fibrotic portion of the injured tendon that was absent in the native, healthy tissue. This staining highlighted the area of disorganized collagen fibers which were rich in glycoproteins and noticeably different from the native tendon morphology (Figure 7b). Additionally, Masson trichrome stained sections allowed us to confirm the occurrence of ectopic bone formation in the tendon side of the enthesis (Figure 7c).

Measurements of the tendon width revealed that the samples corresponding to the injured tissue at 10 days suffered from a significant enlargement (*p* < 0.05) of the tendon when compared to the native tissue (Table 1). The measurement of the tendon width was not possible in the samples from 1 day after the injury due to the recent creation of the defect, which made a reproducible measurement of the tendon unfeasible.

#### 2.4.2. Immunohistochemistry

Samples from the native and injured tissues were stained for collagen I and collagen II, which are two of the main structural collagens present at the tendon-to-bone enthesis. The native tissue samples showed a strong presence of collagen I at both the tendon and bony sides of the enthesis. The presence of collagen II in the healthy tendon was not verified by the corresponding staining at the tendon side of the enthesis, but rather only at the fibrocartilaginous portion of the enthesis (Figure 8).

Injured samples harvested at day 1 after enthesis injury showed a similar pattern of collagen deposition to that described for the healthy tissue. However, there was a significant presence of recently damaged tissue in the tendon and tendon-to-bone interphase areas that made the identification of the histological features of these samples difficult. This was not the case in the injured samples harvested 10 days after enthesis injury; there, the distribution of collagen I was similar to that observed in the native tissue (that is, a strong presence of collagen I at both sides of the enthesis). Remarkably, intense collagen II staining was observed along the fibrotic portion of the tendon, as well as in the surroundings of the abnormal bony structure previously identified in the tendon portion of the enthesis with the prior safranin O and Masson trichrome staining.

### 2.5. In Situ Hybridization

In situ hybridization (ISH) was performed to visualize the tissue localization of previously identified miRNAs that were dysregulated in the injured enthesis samples. As recommended by the manufacturer, the ISH protocol was optimized using the U6 positive control probe, with a scrambled miRNA probe for a negative control. The use of a scrambled miRNA probe allowed us to confirm that no nonspecific interactions were present in our ISH experiments that could hinder the interpretation of our results.

Two relevant miRNAs were investigated (miR-16-5p and miR-133a-3p) in the native tissue samples, as well as in the sample harvested 10 days after the injury. In the native tissue, the expression of miR-16-5p could not be verified by the in situ hybridization. This was not the case for miR-133a-3p, which was localized mainly in the chondrocytes present at the growth plate of the tibia, while the expression in the tendon portion was barely noticeable except for some scattered positive cells in the tendon portion around 500 µm from the enthesis area. Conversely, in the samples from the injured tissue (10 days post-injury), the expression of both miRNAs was abundant in both the fibrotic portions of the tendon and the surroundings of the ectopic ossification area on the tendon side of the enthesis. The expression patterns in the injured samples of both miR-16-5p and miR-133a-3p highlighted a well-defined boundary that divided the tendon longitudinally in two halves along different morphologies and miRNA expression (Figure 9).

## 3. Discussion

The tendon-to-bone enthesis is a frequent site of injury. It has been estimated that 30% of all musculoskeletal consultations are due to a form of tendon or enthesis injury [41,42]. Most tendon or enthesis injuries can heal without the need for clinical intervention. However, non-surgical treatment is usually followed by surgery, due to improper healing of the injured tendon or tendon-to-bone enthesis [43,44]. Hence, the need for current efforts to elucidate the underlying mechanisms of enthesis healing.

In a recently published review, Chartier et al., summarized the three overlapping steps involved in tendon healing: inflammation, proliferation, and remodeling [45]. These three steps take place while mediated by a balance of extrinsic and intrinsic mechanisms [46,47]. Extrinsic mechanisms are active in the early stages of healing. They involve the infiltration of inflammatory cells and fibroblasts from the surrounding tissue (e.g., synovium, tendon sheath, and paratenon), and are usually associated with an increase in collagen disorganization, high levels of glycosaminoglycans, and increased tendon diameter. The timeline for the intrinsic mechanism is usually delayed by a few days when compared to the extrinsic mechanism, comprising the migration of tenoblasts and tenocytes from within the tendon (mostly from the epitenon and endotenon) toward the wound site [45]. Collagen production by tenocytes and by the cells populating the epitenon usually yields more mature and larger collagen molecules than those produced by the cells migrating from the synovial sheath or the paratenon [48]. Thus, several authors have focused on the comparison between both mechanisms, aiming to answer the question of whether or not one of these two healing patterns might be enough for the adequate restoration of the healthy morphology and biomechanics of the tendon [48,49,50].

With our injury model, we were able to follow the progression of enthesis healing for up to 10 days after the injury. The observed enlargement of the tendon, the collagen disorganization, and the proteoglycan-positive staining in the fibrotic portion at 10 days are in line with the aforementioned characteristics of the extrinsic healing mechanism, which seemed to be predominant in this early stage of healing in our injury model [45,49,50]. Additionally, we observed that as a result of such a healing response, an area suffering from ectopic ossification was present in the tendon portion of the enthesis of the injured samples after 10 days. Both the positive collagen II staining and the presence of chondroid metaplasia surrounding the ossification area indicated that such ossification has an endochondral, rather than intramembranous, origin. These observations support the consensus that most of the ectopic ossification of tendons and tendon-to-bone occur by endochondral ossification [10,51,52].

The occurrence of ectopic ossification in both the tendon and the tendon-to-bone enthesis upon injury has been observed in human and animal models as an undesired effect of the failed regeneration of the native tissue structure [10,53]. Tendon ossification is associated with severe pain and an increased risk of rupture, and although it is commonly seen in clinics, it has been poorly characterized [52]. Some authors have associated the formation of ectopic ossification with the extrinsic migration of inflammatory cells (possibly those derived from bone marrow) to the site of the injury, although the precise mechanisms of ectopic ossification of the tendon and enthesis are far from understood [52,54].

In an attempt to shed light on the intricate mechanisms responsible for the occurrence of fibrosis and ectopic ossification of injured tendon-to-bone entheses, we investigated the expression profile of fibrosis-related miRNAs in the early stages of patellar enthesis healing. MiRNAs have drawn significant attention in recent years as being potential therapeutic targets, signaling mediators, and even biomarkers, due to their ability to regulate a multitude of pathways through their suppressing of expression of their target mRNAs [19,55,56]. Several reports indicate that among the most relevant miRNAs associated with early tendinopathy are the members of the miR-29 family [57,58,59]. It has been demonstrated that the members of this family act as post-transcriptional regulators of collagen, and that they are typically dysregulated upon tendon injury [57]. However, in our enthesis injury model, we only observed dysregulation of one member of this family, the miR-29b-3p, which showed a 3.4-fold increase at 10 days, whereas the expression of miR-29a-3p and miR-29c-3p was similar to that of the native tissue. In addition, miR-29b-3p has been reported to play a role in chondrocyte homeostasis and in repressing *SMAD3* signaling. In this regard, we observed that *SMAD3* was downregulated in the injured samples at both time points of observation. However, the suppression of this gene was strongest at 10 days, when the expression of the miR-29b-3p was the highest. This observation is also in line with the strong upregulation of the miR-16-5p at the same time point, which (similarly to miR-29b-3p) is known to target *SMAD3* in chondrocytes, one of the signature cell populations in the patellar enthesis [36,60]

Another set of interesting miRNAs that resulted in upregulated at 10 days but not at one day in our injured samples include miR-17-5p, miR-133a-3p, and miR-182. Both miR-17-5p and miR-133a-3p were predicted by our IPA analysis to target and inhibit *RUNX2*, which we found to be downregulated several-fold at 10 days. This prediction had been confirmed previously using luciferase reporter assays [40,61]. Moreover, a regulatory role for miR-17-5p in ectopic bone formation in ligaments has been described in patients of ankylosing spondylitis, wherein it targets and suppresses the ankylosis protein homolog (*ANKH*) and enhances the expression of *COL1A1* [40]. Furthermore, the stimulatory effect of miR-17-5p over the expression of *COL1A1* was also described in patients with non-traumatic osteonecrosis, as a consequence of the inhibition of *SMAD7* [62]. Interestingly, *ANKH* and *COL1A1* were upregulated in the day 10 samples in comparison to expression levels in the native tissue, while the protein expression level of collagen I at the later time point was similar to that of the native tissue, but significantly higher than the expression level observed one day after the injury.

When upregulated 10 days after the injury, miRNA-182 was predicted to target *MKX*, and resulted in downregulation in our samples of this timeframe. Different regulatory roles have been described for the miR-182, including cell growth, cancer progression, lymphocyte expansion, and even positive regulation of osteoclastogenesis via the miR-182-PKR-IFN-β pathway [63]. However, to the extent of our knowledge, there is no published experimental evidence for the regulation of *MKX* through miR-182 targeting. Our observations might well serve as a possible indication of such interaction in an in vivo setting, worthy of further investigation.

Of all the miRNAs investigated using our injury model, miR-16-5p and miR-133-3p were two of the most dysregulated, showing downregulation immediately after the injury and strong upregulation at 10 days. These two miRNAs are highly interesting, and especially relevant for our injury model since both of them have been proven to act as anti-fibrotic regulators by inhibiting myofibroblast activation [64,65]. The origin of myofibroblasts during wound healing is unclear [66]. However, they play a critical role in matrix remodeling during healing and have been reported to be involved in the maintenance of scar tissue, whether by actively producing fibrotic ECM or by inhibiting the migration of other cell types [67]. The observed upregulation of these miRNAs (and the strong downregulation at the same time point) in the collagen III protein could well indicate that, after 10 days, the healing process of the injured enthesis has been redirected against fibrosis. This would also be in line with the observed downregulation of the miR-155, known to be a major profibrotic regulator in multiple tissues and pathologies, seen at the same time point [68,69].

Our in situ hybridization experiments revealed that the overexpression of both miR-16-5p and miR-133a-3p was specifically localized in the fibrotic portion of the injured tissue and in the area surrounding the site of ectopic ossification. This finding is particularly relevant when studying such a complex interphase tissue as the tendon-to-bone enthesis since it allows us to pinpoint the exact localization of the cell population responsible for the overexpression of such miRNAs. Our observations did not necessarily address the direct molecular consequences of the dysregulation of miR-16-5p and miR-133a-3p in the early healing response of the injured enthesis. However, the highly specific localization of the overexpression of these miRNAs did highlight a potential antifibrotic role in injured enthesis that has not been previously described. Additionally, such findings could serve as a stepping stone toward the development of potential therapeutic strategies, especially those that might rely on either the enhancement or the fine-tuning of the timing and duration of the overexpression of these antifibrotic miRNAs upon injury, owing to the fibrosis that we found to be strongly present in the injured tissue, based on its morphological appearance after 10 days.

With our study, we aimed to characterize a rat-based patellar injury model during the early stages of healing, paying special attention to the local miRNA expression profile. The enlargement of the tendon at the 10-day time point was in line with the rapid increase in collagen III production immediately after the injury, followed by the later increase in collagen I and collagen II. The regulation of extracellular matrix production is a complex mechanism with multiple players, which include some of the miRNAs that we have investigated in our study [70]. The observed downregulation of miRNAs associated with the inhibition of collagen production (e.g., miR-124-3p and miR-133a-3p) immediately following injury could be connected with the increase in ECM deposition at the earlier recorded time points. Similarly, the upregulation of antifibrotic miRNAs at 10 days (e.g., miR-133a-3p and miR-16-5p), in addition to the normalization of the production of collagen I and collagen III, could mark the beginning of the remodeling phase of the enthesis healing process. However, a more in-depth study would be necessary to establish the precise roles of these microRNA in the process of enthesis healing.

Although the knowledge gained paves the way toward deeper insights into the healing mechanisms of the enthesis, our study is not exempt from limitations. We limited our time points of observation to 1 and 10 days after the injury. Adding a later time point could have been beneficial, to assess the evolution of the fibrosis and the ectopic ossification upon the upregulation of the antifibrotic miRNAs observed at the 10-day time point. Nonetheless, in a previous study, we investigated the healing of the enthesis after 4 and 12 weeks from creating a patellar defect [10]. The nontreated controls showed at both time points clear signs of fibrosis and ectopic bone formation in the tendon and enthesis area. This is evidence that scar formation and tendon ossification were neither effectively avoided nor reverted by the native mechanism of enthesis healing, including the overexpression of antifibrotic miRNAs observed in this study after 10 days. Another constraint is the limited selection of mRNA targets for the enthesis-relevant dysregulated miRNAs whose expression had been validated using qPCR in the injured samples. However, by using IPA analysis, we attempted to cover the most fibrotic-relevant predicted targets for our pool of dysregulated miRNAs. Future studies could focus on investigating the expression of other members of the multitude of potential mRNA targets of dysregulated miRNAs upon enthesis injury, in addition to investigating the validation of the miRNA–mRNA interactions.

## 4. Materials and Methods

### 4.1. Rat Patellar Model

A partial enthesis injury animal model was developed for the purpose of this study. All animal procedures were approved by the Mayo Clinic Institutional Animal Care and Use Committee (protocol #A00006605-22).

Twenty male Fischer 344 rats (Charles River Laboratories, Wilmington, MA, USA) at 16 weeks of age were used. Animals were anesthetized with isoflurane (Piramal Critical Care, Talangana, India) in an induction chamber and had the fur on their right hindlimb shaved and cleaned with povidone iodine (Professional Disposables Internationals, Orangeburg, NY, USA) and 70% ethanol. All animals received a single dose of subcutaneous cefazolin (50 mg/kg; antibiotic (Pharmaceuticals, Columbus, OH, USA) and slow-release buprenorphine (1 mg/kg; for analgesia (Zoopharma, Laramie, WY, USA) prior to surgery. Animals were then placed on the surgical table in the dorsal recumbency position and the right hindlimb was left exposed using a sterile fenestrated drape. All rats received a unilateral, partial patellar enthesis injury in their right limb, and their left patellar enthesis was used as native control. A 5 mm medial parapatellar incision was swiftly created on the right knee, using a #15 scalpel blade. The patellar enthesis was exposed and a 2 mm injury was created on the medial aspect using a #11 scalpel blade in the axial plane. The skin was then closed using one 9 mm wound clip and the animal was transferred to a heated recovery chamber, set at 32 °C. Once the animal recovered from anesthesia, it was transferred to its cage where it was allowed to bear weight and access to food and water ad libitum.

At the day 1 and day 10 (*n =* 10 respectively) time points, the animals were euthanized using an automated CO_2_ delivery system. The injured and contralateral patellar enthesis were then harvested for analysis. The knee joint was accessed above the patella and the anterior cruciate ligament tendon was then cut, allowing for anterior displacement of the tibia. A coronal cut was created on the tibial plateau, posterior to the enthesis and an axial cut on the metaphysis, distal to the tibial tuberosity using a rotary saw blade. Samples undergoing histological analysis were immediately processed. Samples undergoing PCR analysis were placed in 5 mL of RNAlater (Invitrogen, Waltham, MA, USA) and subsequently trimmed closer to the enthesis.

### 4.2. Sample Preparation and RNA Extraction

Samples from the injured patellar entheses tissue from both time points of observation (*n* = 5 per time point), in addition to the contralateral/native tissues (*n* = 5), were recovered from RNAlater, snap-frozen in liquid nitrogen, and homogenized with steel beads in the presence of TRIzol (SigmaAldrich, St. Louis, MO, USA) using a TissueLyser II set to cycles of 3 min at 30 Hz (Qiagen GmbH, Hilden, Germany). The isolation of the total RNA was performed following the well-established phenol-chloroform extraction protocol. The concentration and purity of the extracted RNA were measured using a NanoDrop spectrophotometer (NanoDrop Tech. Inc., Greenville, SC, USA). In every case, the ratios 260/230 and 260/280 were found to be ≥1.8.

### 4.3. cDNA Synthesis and PCR Array for miRNA Expression

The extracted total RNA was the starting material for the cDNA synthesis reactions. Then, cDNA synthesis was performed using the miScript II RT Kit (Qiagen GmbH, Hilden, Germany) in a C1000 Touch Thermal Cycler (Eppendorf AG, Hamburg, Germany) following the instructions from the manufacturer. To ensure the PCR quantification of mature miRNAs only, the cDNA synthesis mix was prepared using the miScript HiSpec buffer. Afterwards, cDNA samples from native tissue taken from both time points were pooled and used as the template for the qPCR array. The qPCR array was performed in a CFX 96 Real-Time System thermocycler (Bio-Rad, Hercules, CA, USA) utilizing a Fibrosis-Pathway focused miScript qPCR array (MIRN-117Z) and the miScript SYBER^®^ Green PCR (Qiagen GmbH, Hilden, Germany).

Following Qiagen’s instructions, the obtained Ct values from the qPCR array were exported to an Excel file and uploaded to Qiagen’s data analysis web portal http://www.qiagen.com/geneglobe, accessed on 13 July 2022. Samples were assigned to both control (native tissue) and test groups (injury 1 day, injury 10 days). The Ct values were normalized against the arithmetic mean of the Ct values from the reference genes included in the array (i.e., Snord61, Snord68, Snord72, Snord95, Snord96A, Rnu6-6p). For each time point of observation, the fold regulation (FR) of each miRNA was calculated with respect to the native tissue, following the 2^(−∆∆Ct)^ method for fold change. Treated data have been displayed as fold regulation (FR), as this facilitates the interpretation of the results. FR is the same as fold change (FC) when FC ≥ 1, and FR is the inverse negative of the FC when FC < 1.

Additionally, qPCR was performed on the individual enthesis samples for the relevant miRNAs that were shown to be dysregulated by the qPCR array. For this, the SYBR^®^ Green-based PCR miScript Primer assay for each respective miRNA to validate was used (Qiagen GmbH, Hilden, Germany) and the expression was normalized against the reference miRNA Snord68.

### 4.4. cDNA Synthesis and PCR for mRNA Expression

Total RNA was used as starting material for the cDNA synthesis reaction using the iScript^TM^ cDNA synthesis kit (Bio-Rad, Hercules, CA, USA) and the C1000 Touch Thermal Cycler. The qPCR reaction mix was prepared with the iQ^TM^ SYBR^®^ Green supermix (Bio-Rad, Hercules, CA, USA) following the manufacturer’s instructions. A list of the primers used can be found in Table 2. Gene expression was normalized against the reference gene β-tubulin, and the fold regulation of the expression was calculated in the same manner as previously described.

### 4.5. Ingenuity Pathway Analysis

The identities of those miRNAs dysregulated by more than twofold in the injured samples (compared to the native tissue) were uploaded to the Ingenuity Pathway Analysis software v012004 (IPA, Qiagen GmbH, Hilden, Germany). Then, the miRNA target filter tool was employed to select miRNAs with mRNA targets either known or predicted to be associated with diseases and/or functions relevant to enthesis healing or injury. The filter parameters used were (i) species/tissue: rat/tendon or rat/cartilage, (ii) cell type: tenocyte, chondrocyte, or osteoblast, (iii) confidence: experimentally observed or highly predicted, and (iv) diseases and functions: tendon tissue or cartilage tissue.

In addition to the target prediction from the IPA software v012004, we performed extensive literature research to identify targets that were not yet included in the IPA knowledge database.

### 4.6. Protein Purification and Western Blots

After total RNA extraction with the TRIzol reagent, the fraction of total protein was isolated and solubilized following a standard chloroform/isopropanol/guanidine hydrochloride extraction protocol [71]. RNA and DNA were swiftly extracted, and isopropanol (SigmaAldrich, St. Louis, MO, USA) was added to the sample to precipitate the proteins. After centrifugation, the pellet was washed in guanidine hydrochloride 0.3 M (SigmaAldrich, St. Louis, MO, USA), followed by a final wash with ethanol. After air-drying, the pellet was resuspended in SDS 1% (SigmaAldrich, St. Louis, MO, USA). For Western blotting, the total protein content was quantified using the bicinchoninic acid (BCA) method, utilizing the Pierce^TM^ BCA kit (ThermoFisher, Waltham, MA, USA), and 15 µg were loaded in each well of 8% bis-acrylamide gel. Electrophoresis was performed in a running buffer (Tris, SDS, Glycine) for 60 min at 120 V. After separation, proteins were transferred on a nitrocellulose membrane using transfer buffer (Tris-base, glycine, SDS, methanol) under 350 mA for 90 min. Ponceau S staining (SigmaAldrich, St. Louis, MO, USA) was performed and membranes were then blocked for 60 min at room temperature in a solution of 5% non-fat dry milk (Merck KGaA, Darmstadt, Germany), diluted in Tris Buffer Saline supplemented with 0.1% Tween20 (TBST (Merck KGaA, Darmstadt, Germany)). Primary antibodies used for immunodetection were incubated overnight at 4 °C: collagen I (1/1000; ab270993), collagen II (1/2000; ab34712), collagen III (1/1000; ab6310), and collagen X (1/500; 2031501005). Primary antibodies for collagen I, collagen II, and collagen III were purchased from Abcam (Cambridge, UK), and anti-collagen X was purchased from Quartett (Berlin, Germany). After rinsing, the membranes were incubated with horseradish peroxidase (HRP)-conjugated secondary antibodies (1/3000) at room temperature for 60 min. All antibodies were diluted in TBST/milk 5%. The protein signal was developed using a Clarity Western ECL substrate (Bio-Rad, Hercules, CA, USA) and chemiluminescence was detected using Chemidoc technology (Bio-Rad, Hercules, CA, USA). The intensity of the band was then quantified using ImageJ software 1.53t (NIH, Bethesda, MD, USA) and normalized to total proteins (Ponceau S).

### 4.7. Histology

The histological characterization included safranin O, alcian blue, and Masson trichrome staining. Additionally, immunohistochemical staining (IHC) for collagen I and collagen II was performed.

The samples used for histology (*n =* 5 per group) were fixed for 48 h using 4% paraformaldehyde (SigmaAldrich, St. Louis, MO, USA). Subsequently, the samples were rinsed with PBS (Thermo Fisher Scientific, Landsmeer, the Netherlands), and decalcified for 30 days using 10% buffered EDTA solution (SigmaAldrich, St. Louis, MO, USA) with regular buffer exchange every two to three days. The endpoint of the decalcification was determined by X-ray.

After the decalcification, the samples were dehydrated in an increasingly concentrated ethanol series and embedded in paraffin. Samples were sectioned to 7 microns using a Leica RM 2165 microtome (Leica Biosystems, Nussloch, Germany).

Before each staining, the samples were rehydrated in descending strength ethanol series in combination with distilled water, after two changes of NeoClear-xylene substitute (Merck KGaA, Darmstadt, Germany).

For the safranin O staining, the rehydrated samples were incubated for 10 min with hematoxylin solution (Carl Roth GmbH, Karlsruhe, Germany), followed by 5 min staining with 0.1% fast green solution (SigmaAldrich, St. Louis, MO, USA). Afterward, the samples were rinsed with 0.1% acetic acid, and further stained with 0.1% safranin O solution for 10 min (SigmaAldrich, St. Louis, MO, USA). Thereafter, the samples were dehydrated in ascending ethanol series, cleared with NeoClear-xylene substitute, and mounted with UltraKit mounting media (Thermo Fisher Scientific, Landsmeer, The Netherlands). The Alcian blue staining was performed by incubating the rehydrated slides in a working solution of alcian blue at pH 2.5 (SigmaAldrich, St. Louis, MO, USA) for 10 min, followed by three washes with distilled water. Subsequently, samples were rehydrated, cleared, and mounted as described above.

For the Masson trichrome staining, the Trichrome Stain (Masson) kit was purchased from SigmaAldrich (St. Louis, MO, USA). The staining was performed following the indications from the manufacturer. Briefly, the samples were incubated for 15 min in preheated Bouin’s Solution at 56 °C. Later, the slides were washed in tap water to remove the yellow color from the section. Consequently, the slides were stained for 5 min in hematoxylin solution (Carl Roth GmbH, Karlsruhe, Germany), washed in running tap water for 5 min, stained in Biebrich scarlet-acid fuchsin for 5 min, and rinsed with deionized water. Finally, the slides were placed in a working phosphotungstic/phosphomolybdic acid solution for 5 min, followed by a 5 min incubation in aniline blue solution, and a 2 min incubation with 1% acetic acid. Once the protocol was completed, the samples were rehydrated, cleared, and mounted as described above.

For the IHC staining, the primary and secondary antibodies (Table 3) and DAPI counterstaining were purchased from Abcam (Cambridge, UK).

Antigen retrieval was conducted using 10 mM citrate buffer (pH 6 (Abcam, Cambridge, UK) for 10 min at 95 °C. Subsequently, blocking was carried out by incubating the samples with blocking solution (0.1% Triton, 1% BSA, 5% goat serum) (SigmaAldrich, St. Louis, MO, USA) for 1 h at room temperature. Afterward, the slides were placed in a humidity chamber and incubated overnight at 4 °C, with the primary antibody diluted in blocking solution without goat serum. This was followed by incubation with the secondary antibody for 2 h at room temperature. Finally, counterstaining with DAPI was performed and the slides were mounted with Dako fluorescent mounting media (Agilent Technology, Santa Clara, CA, USA).

The stained slides were visualized using a Nikon DS-Ri2 camera mounted on a Nikon Ti Slide Scanner Microscope (Nikon Instruments Europe BV, Amsterdam, The Netherlands).

Histology images of the native tissue and the injured tissue corresponding to time point 10 days were uploaded to Image J v1.53p (NIH, Bethesda, MD, USA) The width of the tendon was measured in the samples using the measurement tool of Image J. The measurements were imported into GraphPad Prism 8.0 (GraphPad Software, San Diego, CA, USA). Statistically significant differences were determined by an unpaired *t*-test with Welch’s correction *p* < 0.05.

### 4.8. In Situ Hybridization

For the in situ hybridization (ISH) experiments, the double-DIG miRCURRY LNA detection probe, miRCURRY ISH Buffer, and control probes were purchased from Qiagen (Qiagen GmbH, Hilden, Germany). The sheep anti-DIG-AP, sheep serum, NBT/BCIP ready-to-use tablets, levamisole hydrochloride, and the 30% BSA solution were purchased from Sigma-Aldrich (St. Louis, MO, USA) and the ultrapure SSC 20× buffer was purchased from Thermo Fisher Scientific (Thermo Fisher Scientific Inc., Waltham, MA, USA). The hybridization protocol was conducted as recommended by Qiagen. All steps of the ISH experiments were conducted in RNase-free conditions, following the manufacturer’s recommendations.

Briefly, the samples were deparaffinized and rehydrated in NeoClear-xylene substitute, descending serial dilutions of ethanol, and RNase-free distilled water (Qiagen GmbH, Hilden, Germany). Subsequently, proteinase K (SigmaAldrich, St. Louis, MO, USA) was added to the slides, and the samples were incubated for 10 min at 37 °C using a Dako hybridizer (DAKO Colorado, Inc., Fort Collins, CO, USA). Afterward, the slides were washed twice with sterile PBS (Thermo Fisher Scientific Inc., Waltham, MA, USA) and incubated with the hybridization mix containing the detection probes or controls, respectively, for 60 min at 55°C in the Dako hybridizer. The hybridization probes were used the recommended concentrations of 1 nM for the LNA U6 snRNA detection probe and 40 nM for the double-DIG miRNA and the scramble detection probes.

This hybridization step was followed by stringent washes with 5×, 1×, and 0.2× SSC buffer at 55 °C. Blocking was performed for 15 min at room temperature before a 1 h incubation with the anti-DIG (dilution 1:800). The slides were then washed with PBS, and a freshly prepared AP substrate was added to the samples, which were then incubated for 2 h at 30 °C in a humidity chamber. The AP reaction was stopped by incubating the samples two times for 5 min each with KTBT buffer (50 mM TrisHCl, 150 mM NaCl, 10 mM KCl). Subsequently, the samples were counterstained with fast green solution (Carl Roth GmbH, Karlsruhe, Germany) for 4 min, washed for 10 s with a 1% solution of acetic acid and water. Finally, the samples were dehydrated in an increasingly concentrated ethanol series before mounting with UltraKit mounting media (J.T. Baker, Leicestershire, UK). Images were taken with a Nikon DS-Ri2 camera, mounted on a Nikon Ti Slide Scanner Microscope (Nikon Europe B.V., Amstelveen, The Netherlands).

## 5. Conclusions

Tendon and enthesis injuries are debilitating conditions that affect a growing number of people worldwide. Here, we have characterized the early healing response of an injured patellar enthesis in a rodent model. We described the occurrence of fibrotic scar tissue 10 days after the creation of the injury, in addition to the occurrence of ectopic ossification at the tendon-to-bone enthesis. Moreover, we reported the dysregulated expression of at least 13 enthesis-relevant miRNAs both 1 day and 10 days after the injury, whose predicted mRNA targets are known to play relevant roles in the process of enthesis healing and regeneration. Additionally, we were able to localize the expression of miR-16-5p and miR-133a-3p in the fibrotic portion of the injured tissue, which indicates a direct relationship between the overexpression of such miRNAs and the healing of the tendon-to-bone enthesis. We believe that the results described in the present manuscript could stimulate future studies to explore the therapeutic potential of such miRNAs by their ability to tune the expression of their mRNA targets, and thereby improve the regeneration of injured entheses.

## Figures and Tables

**Figure 1 ijms-24-08556-f001:**
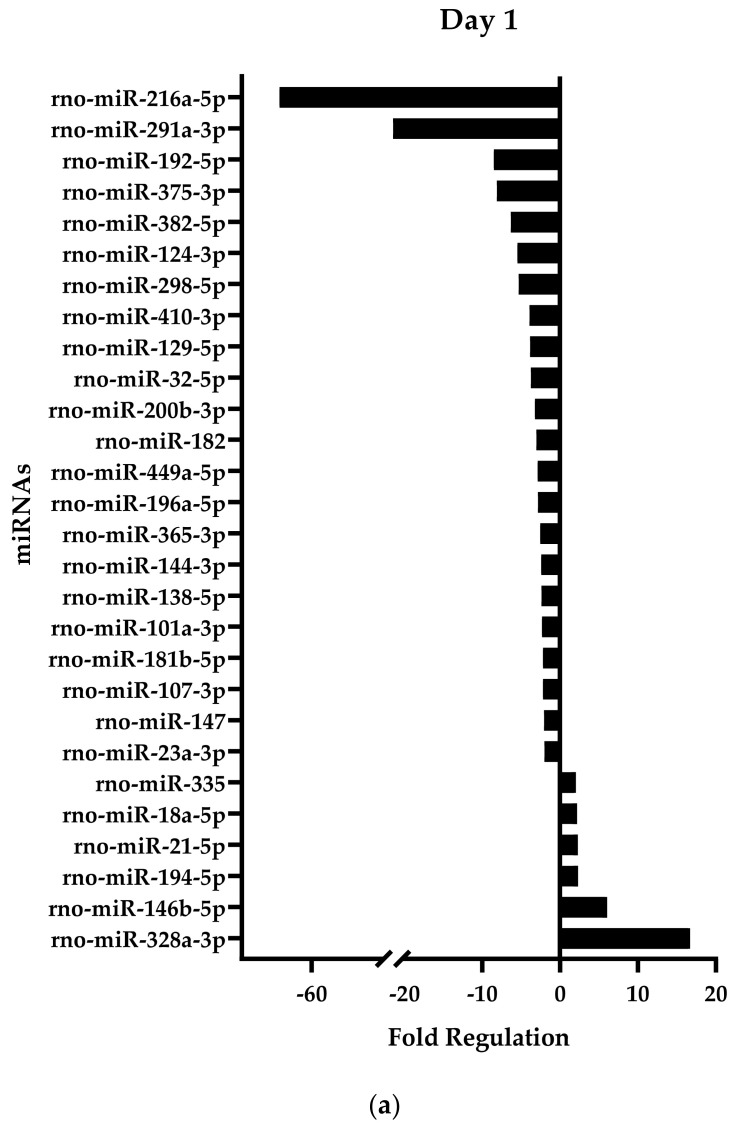
Dysregulated miRNAs following injury at (**a**) day 1, and (**b**) day 10, with respect to the expression in the native tissue.

**Figure 2 ijms-24-08556-f002:**
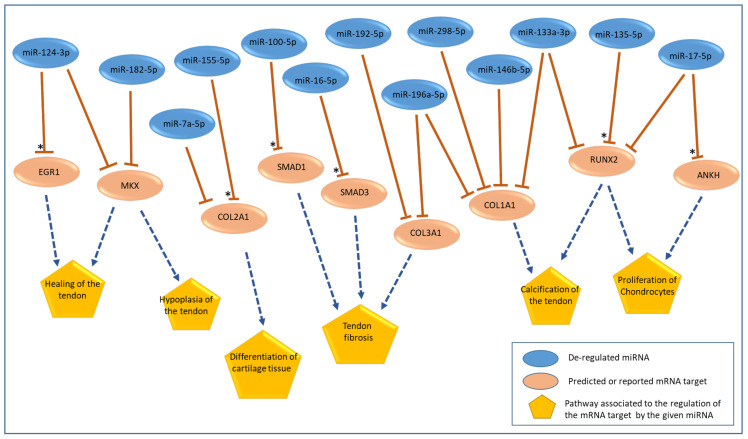
Schematic representation of the interactions between the aberrantly regulated miRNAs at early stages of healing on an enthesis injury in the rats’ patellae. The predicted and/or reported mRNA targets in the context of tendon- and cartilage-relevant pathways are indicated. The asterisk symbol (*) indicates interactions between the deregulated miRNAs and mRNA target genes found therein that had not been predicted by the IPA, but which have been reported in the literature as targets of the respective miRNA [36,37,38,39,40].

**Figure 3 ijms-24-08556-f003:**
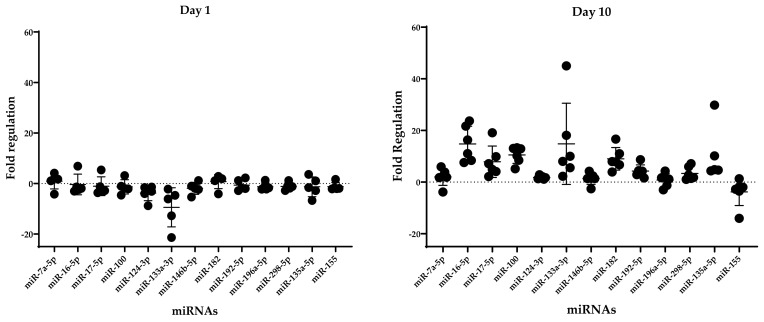
Fold regulation (Mean ± SD) of the expression of the deregulated miRNAs in the injured samples, with respect to the native tissue.

**Figure 4 ijms-24-08556-f004:**
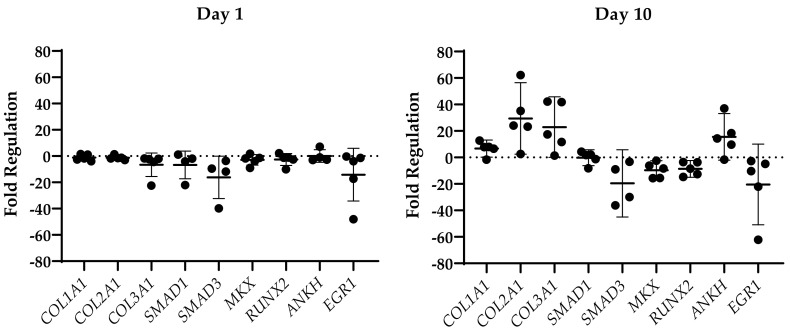
Fold regulation (Mean ± SD) of the expression of the targets mRNA in the injured samples with respect to the native tissue.

**Figure 5 ijms-24-08556-f005:**
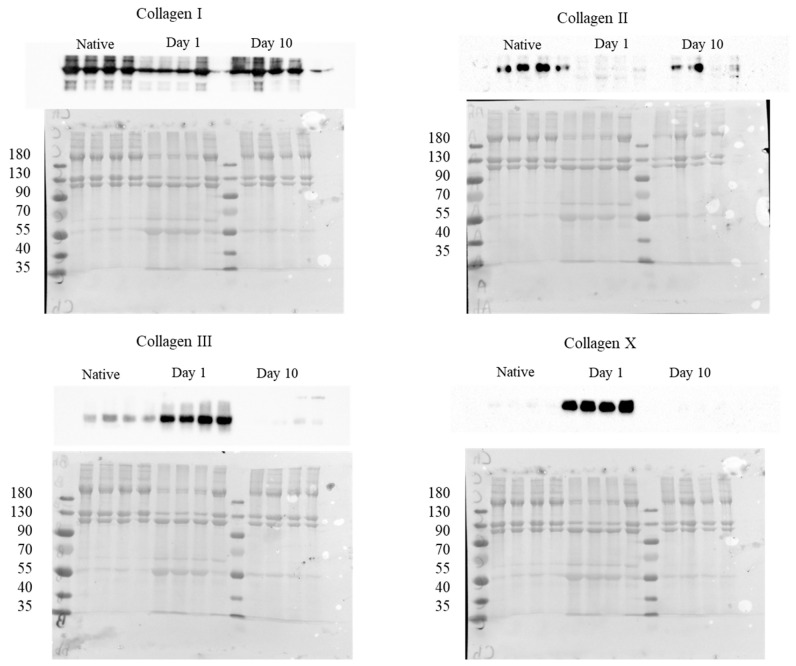
Western blot for proteins of the native tissue samples, with both injured tissue at 1 day and at 10 days following injury. Each lane corresponds to an individual sample. Normalized against total protein content.

**Figure 6 ijms-24-08556-f006:**
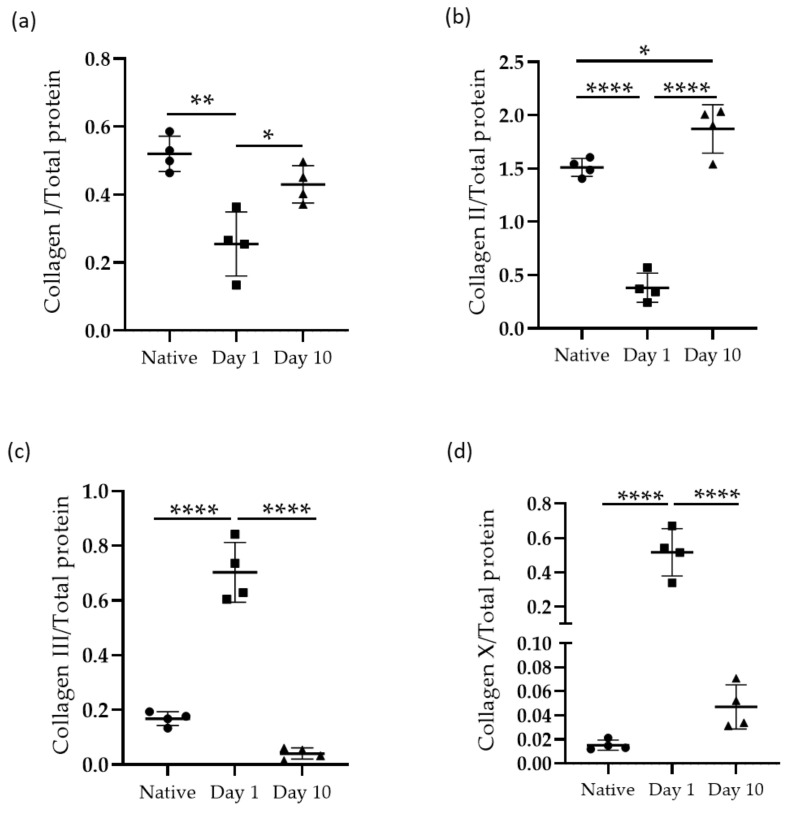
Protein expression of collagen I (**a**), collagen II (**b**), collagen III (**c**) and collagen X (**d**) normalized against total protein content. Results are illustrated for both the native and injured tissue samples at each time point of observation (Mean ± SD). Statistical significance is indicated by * *p* < 0.05, ** *p* < 0.01, and **** *p* < 0.0001.

**Figure 7 ijms-24-08556-f007:**
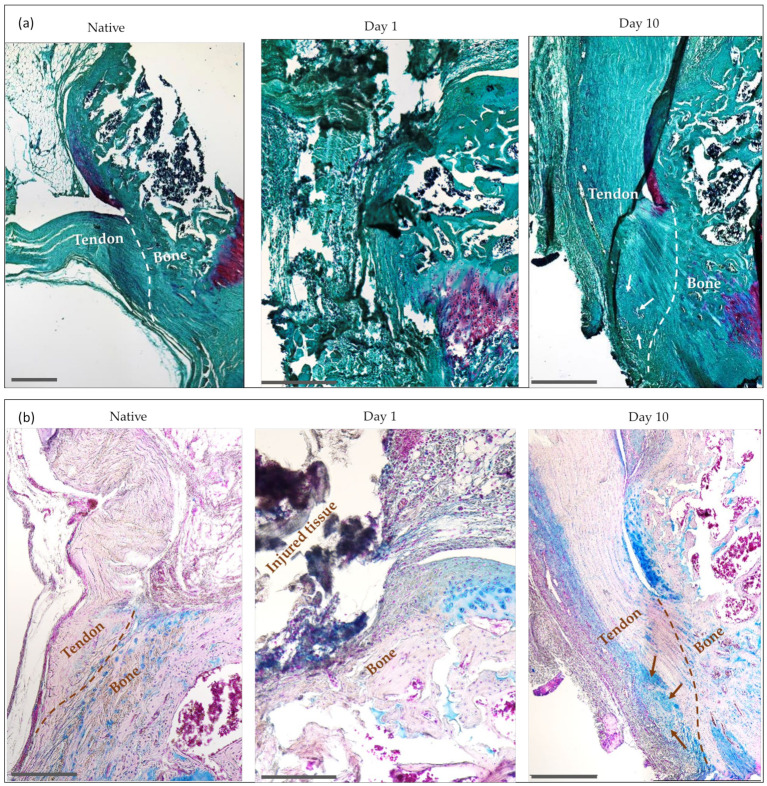
Safranin O (**a**), alcian blue (**b**) and Masson trichrome staining (**c**) of both native and injured samples (1 day and 10 days after the injury). Dashed lines indicate the tendon-to-bone interphase or enthesis. Arrows indicate ossification in the tendon side of the enthesis. Scale bar = 500 µm.

**Figure 8 ijms-24-08556-f008:**
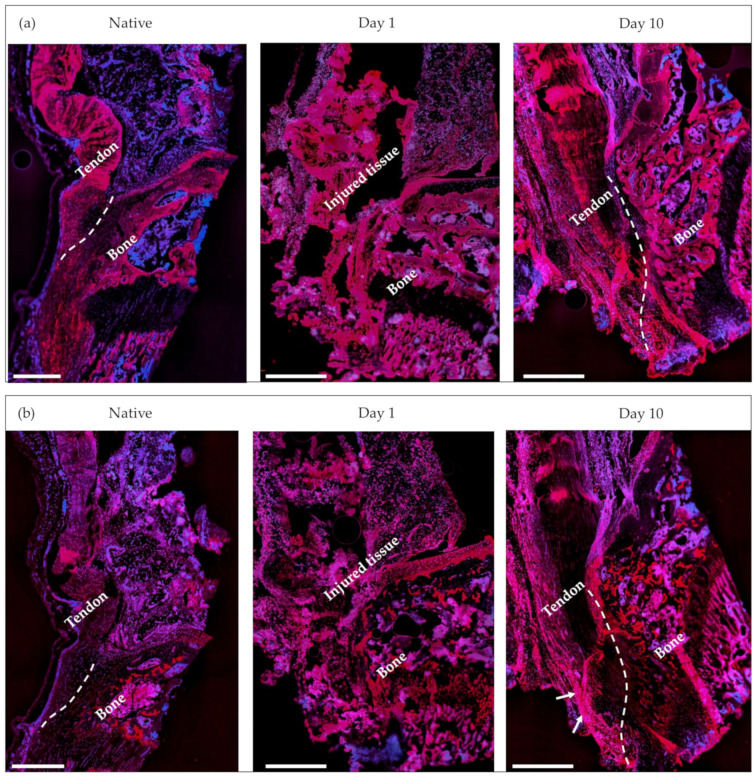
Collagen I (**a**) and collagen II (**b**) staining of native and injured tissue samples (1 day and 10 days after enthesis injury). Dashed lines indicate the tendon-to-bone interphase. Red = collagen, blue = DAPI. Arrows indicate ossification in the tendon side of the enthesis. Scale bar = 500 µm.

**Figure 9 ijms-24-08556-f009:**
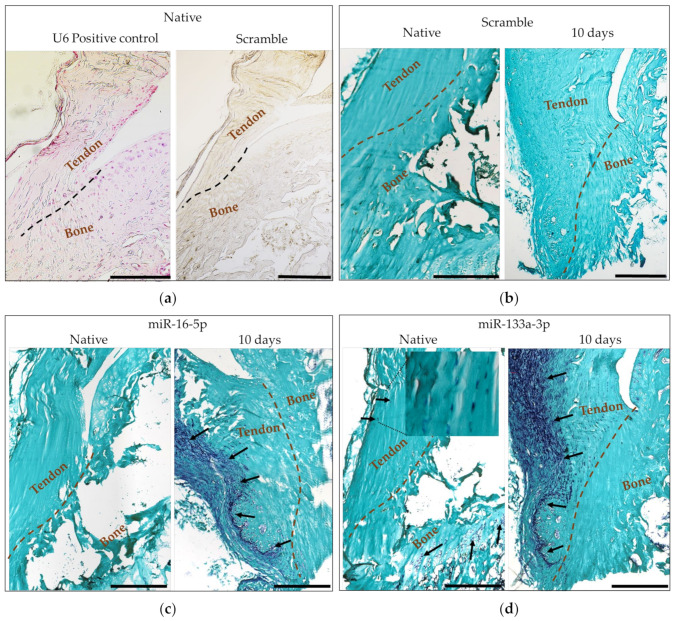
In situ hybridization. U6 positive control and scramble negative control in native tissue (**a**); Scramble negative control in native and injured tissue 10 days after injury (**b**); and miR-16-5p (**c**) and miR-133a-3p (**d**) in both native and injured tissue 10 days after the injury. Dashed lines indicate the tendon-to-bone interphase. Arrows indicate the presence of the corresponding miRNA: dark blue = miRNA expression, and light green = counterstaining with fast green. Scale bar = 500µm.

**Table 1 ijms-24-08556-t001:** Tendon width (Mean + SD) of the native tissue and the injured tissue harvested at 10 days after enthesis injury.

Samples	Tendon Width (µm)
Native tissue	341.0 ± 78.4
Injured tissue (time point: 10 days)	807.6 ± 31.8

**Table 2 ijms-24-08556-t002:** Primers used for the qPCR reactions for the mRNA targets.

Targets	Forward 5′ *→* *3′*	Reverse 5′ *→* *3′*
*COL1A1*	TTTCCCCCAACCCTGGAAAC	CAGTGGGCAGAAAGGGACTT
*COL2A1*	CACGCCTTCCCATTGTTGAC	AGATAGTTCCTGTCTCCGCCT
*COL3A1*	TGCAATGTGGGACCTGGTTT	GGGCAGTCTAGTGGCTCATC
*MKX*	GACGACGGCTGAAGAACACTG	CCTCTTCGTTCATGTGAGTTCTTGG
*RUNX2*	CAAGGAGGCCCTGGTGTTTA	AAGAGGCTGTTTGACGCCAT
*SMAD1*	CAATAGAGGAGATGTTCAAGCAGT	CAGACCGTGGTGGGATGAAA
*SMAD3*	CTGGTGCTGGGGTTAGGTCA	GGCCATCCAGGGACTCAAAC
*EGR1*	GCACCCACCTTTCCTACTCC	GTGTAAGCTCATCCGAGCGA
*ANKH*	CTGGTGGGATGTGCCTCAAT	GACCGTGTTGTTCGTGTTGG
*TUBB*	GAGGGCGAGGACGAGGCTTA	TCTAACAGAGGCAAAACTGAGCACC

**Table 3 ijms-24-08556-t003:** List of antibodies and working dilutions used in IHC.

	*Abcam ID Number*	*Working Dilution*
*Anti-collagen type I*	ab270993	1/250
*Anti-collagen type II*	ab34712	1/50
*Rabbit IgG isotype control*	ab172730	1/50
*Alexa Fluor 647 Goat anti Rabbit*	ab190565	1/500

## Data Availability

The datasets generated and analyzed during the current study are available from the corresponding author on reasonable request.

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
