# Peer review of "MiRNAs as Potential Regulators of Enthesis Healing: Findings in a Rodent Injury Model"

_ijms, 2023, doi:10.3390/ijms24108556_

Round 1
Reviewer 1 Report
This is an interesting approach of a relevant problem of enthesis repair. The authors apply a rodent model of a frequently human injury. Although the study is admittedly initial it presents interesting data worthwhile of further investigation.
Major points:
1. How can it be sure that increase of micro RNA levels does mean that the possible target level should be decreased? Is not it possible that changes of inhibitor and target could be a parallel in control of gene expression? The relevance of positive or negative correlations should be better supported.
2. The control of in situ hybridization is missing. Gene expression at healing sites could result in nonspecific background. Scrambled oligo probe should be used.
Author Response
Comments from Reviewer 1:
This is an interesting approach of a relevant problem of enthesis repair. The authors apply a rodent model of a frequently human injury. Although the study is admittedly initial it presents interesting data worthwhile of further investigation.
Major points:
- How can it be sure that increase of micro RNA levels does mean that the possible target level should be decreased? Is not it possible that changes of inhibitor and target could be a parallel in control of gene expression? The relevance of positive or negative correlations should be better supported.
- The control of in situ hybridization is missing. Gene expression at healing sites could result in nonspecific background. Scrambled oligo probe should be used.
Our answer to reviewer 1:
Point 1.
As the reviewer pointed out, it is indeed not possible to establish a direct relationship between the up/down-regulation of miRNAs and the down/up-regulation of mRNA merely on the observation of their expression levels. For this reason, we only investigated the expression of mRNAs that were either mathematically predicted or described in the published literature to be targets of the inhere found dysregulated miRNAs. Hence, the interactions that are inferred during the discussion of our results don’t stand unfounded. Furthermore, our study is descriptive and with it, we aim to bring attention to the potential significance of such predicted/reported miRNA-mRNA interactions in the context of enthesis healing, which, to our knowledge, has never been done before. For example, in our study, we found Runx2 to be downregulated in the injured samples while miR-133a was upregulated at the same time point. This miRNAs was predicted by the IPA software to target Runx2 in rat tissue. Additionally, it has been also proven by luciferase assay that miR-133a directly targets RUNX2 in human vascular smooth muscle cells. Based on such evidence, our observations suggest that the downregulation of Runx2 in our injured samples could be related to the upregulation of miR-133a. We understand that the results described in our manuscript, in particular regarding the potential roles of miRNAs as regulators of enthesis-relevant mRNA targets during healing might not definitive, but are undoubtedly a stepping stone for future research focused on the regulation of miRNAs/mRNA for better enthesis healing.
We have re-written a part of our discussion to better support the miRNA-mRNA interactions that are described in our study.
Point 2
As mentioned by the reviewer, the use of scramble oligo probes (miRNA with non-specific sequence) during the in situ hybridization experiments (ISH) is necessary to establish that our stainings are free from non-specific background. As we mentioned in Materials and Methods, we used the scramble miRNA probe included in the “ISH buffer and control kit” purchased from Qiagen as our negative control but we did not include the images in the original version of the manuscript. Following the recommendations from the reviewer, we have added the staining with the scramble probes in the main body of the manuscript.

Reviewer 2 Report
This article has described the miRNA expression profile presented in enthesis tissue upon injury. Authors listed the expression of miRNA at days 1 and 10 after injury and compared with the native tissue, also talked about the levels of collagens by western blot. Safranin O, Alcian blue and Masson trichrome staining also applied for injured tissue and native tissue. My overall impression is that the authors gave a lot of description for different methods and results, but hard to connect them and lack a conclusion or the direction for next step.
The introduction only gave the general information for the relation between mRNA and mMRA, since in the following results, several different miRNAs have been described and compared, but there is no detail for those individual miRNA and the related mRNA (or which biological process this miRNA can be involved in), if we add more detail for this, it would be more clear why we should focus on the expression of those miRNA.
Another main problem is that we can get several results from different experiment, but we don’t have a clear conclusion what we can know from them also the connection of those results. After getting those results for the miRNA dysregulated expression after the injury to the width of tendon or the protein levels of collagens, what those result reflected, also is there any existing experiments showing similar results?
Otherwise, all the figures and tables were clear and explained well. All the methods were in detail and clear.
Author Response
Comments from Reviewer 2:
This article has described the miRNA expression profile presented in enthesis tissue upon injury. Authors listed the expression of miRNA at days 1 and 10 after injury and compared with the native tissue, also talked about the levels of collagens by western blot. Safranin O, Alcian blue and Masson trichrome staining also applied for injured tissue and native tissue. My overall impression is that the authors gave a lot of description for different methods and results, but hard to connect them and lack a conclusion or the direction for next step.
The introduction only gave the general information for the relation between mRNA and mMRA, since in the following results, several different miRNAs have been described and compared, but there is no detail for those individual miRNA and the related mRNA (or which biological process this miRNA can be invo in), if we add more detail for this, it would be more clear why we should focus on the expression of those miRNA.
Another main problem is that we can get several results from different experiment, but we don’t have a clear conclusion what we can know from them also the connection of those results. After getting those results for the miRNA dysregulated expression after the injury to the width of tendon or the protein levels of collagens, what those result reflected, also is there any existing experiments showing similar results?
Otherwise, all the figures and tables were clear and explained well. All the methods were in detail and clear.
Our answer to reviewer 2
As suggested by the reviewer, we have included a short paragraph in the introduction of our manuscript to further clarify the relevance of miRNAs in the process of enthesis healing and why we are interested in investigating their role in enthesis fibrosis. Similarly, we have added a more conclusive summary at the end of our discussion to better clarify the connection between our various experiments. However, it is important to mention that, to the extent of our knowledge, little is known about the roles of miRNAs in the context of enthesis healing. Hence, the character of our study is descriptive. Nevertheless, most of our observations regarding miRNAs, gene, and protein expression are supported by our analysis based on mathematical predictions (IPA software) and/or reports from published literature where similar miRNA-mRNA interactions have been described in other tissues or cell types.

Round 2
Reviewer 1 Report
The authors responded to the major comments.